# *Ginkgo biloba* extract inhibits hippocampal neuronal injury caused by mitochondrial oxidative stress in a rat model of Alzheimer's disease

**Chenyi Xia**[1]*, **Mingmei Zhou**[2], **Xianwen Dong**[3], **Yan Zhao**[1], **Meifang Jiang**[4], **Guoqin Zhu**[4]*, **Zhixiong Zhang**[1]*

**1** School of Integrative Medicine, Shanghai University of Traditional Chinese Medicine, Shanghai, China, **2** Institute of Interdisciplinary Integrative Medicine Research, Shanghai University of Traditional Chinese Medicine, Shanghai, China, **3** Department of Children Rehabilitation Medicine, The Fifth Affiliated Hospital of Zhengzhou University, Zhengzhou, China, **4** SPH XingLing Sci. & Tech. Pharmaceutical Co., Ltd., Shanghai, China

* yushengxcy@163.com (CX); zhuguoqin@xingling.com.cn (GZ); zhixiongzh@163.com (ZZ)

**Data Availability Statement:** All relevant data are within the manuscript and its Supporting Information files.

## Abstract

*Ginkgo biloba* extracts (GBE) have been shown to effectively improve cognitive function in patients with Alzheimer's disease (AD). One potential therapeutic strategy for AD is to prevent loss of adult hippocampal neurons. While recent studies have reported that GBE protects against oxidative stress in neurons, the underlying mechanisms remain unclear. In this study, an AD-like rat model was established via bidirectional injection of amyloid beta 25–35 ($A\beta_{25-35}$; 20 μg) in the hippocampal CA1 region. Learning and memory abilities of experimental rats were AD assessed in response to oral administration of 7.5 g/L or 15 g/L *Ginkgo biloba* extract 50 (GBE50) solution and the peroxidation phenomenon of hippocampal mitochondria determined via analysis of mitochondrial $H_2O_2$ and several related enzymes. Levels of the oxidative stress-related signaling factor cytochrome C (Cyto C), apoptosis-related proteins (Bax, Bcl-2 and caspase-3) and caspase-activated DNase (CAD) were further detected via western blot. 8-Hydroxydeoxyguanosine (8-OHdG), the major product of DNA oxidative stress, was evaluated to analyze DNA status. Our results showed elevated $H_2O_2$ levels and monoamine oxidase (MAO) activity, and conversely, a decrease in the activities of superoxide dismutase (SOD) and glutathione peroxidase (GSH-Px) in the hippocampus of AD rats. Administration of GBE50 regulated the activities of these three enzymes and induced a decrease in $H_2O_2$. GBE50 exerted regulatory effects on abnormally expressed apoptotic proteins in the AD rat hippocampus, enhancing the expression of Bcl-2, inhibiting release of Cyto C from mitochondria, and suppressing the level of caspase-3 (excluding cleaved caspase-3). Furthermore, GBE50 inhibited DNA damage by lowering the generation of 8-OHdG rather than influencing expression of CAD. The collective findings support a protective role of GBE50 in hippocampal neurons of AD-like animals against mitochondrial oxidative stress.

**Funding:** This work was supported by the budget of Shanghai University of Traditional Chinese Medicine (No.2013JW02). The funders had no role in study design, data collection and analysis, decision to publish, or preparation of the manuscript.

**Competing interests:** Chenyi Xia declares no conflicts of interest. Zhixiong Zhang has received a research grant from SPH Xing Ling Sci. & Tech. Pharmaceutical Co., Ltd. (Shanghai, China). Meifang Jiang and Guoqin Zhu are employees of the above company. The remaining authors have no conflicts of interest to declare. This does not alter our adherence to PLOS ONE policies on sharing data and materials. There are no patents, products in development or marketed products associated with this research to declare.

## 1. Introduction

Alzheimer's disease (AD) is a common neurodegenerative disorder that causes progressive cognitive dysfunction and memory impairment. AD is characterized by neuronal loss, extracellular deposits of amyloid beta (Aβ) in senile plaques and intraneuronal neurofibrillary tangles [1,2]. In the brain of AD patients, Aβ deposits constituting senile plaques are believed to play a key role in oxidative stress-mediated pathogenesis [3]. Several studies have shown that Aβ triggers the occurrence of oxidative stress, which, in turn, promotes further accumulation and deposition of Aβ [4–6]. Accumulated Aβ aggregates have been shown to play a pivotal role in oxidative stress, leading to mitochondrial dysfunction, energy failure, lipid peroxidation, protein oxidation and DNA/RNA oxidation that contribute to the pathogenesis of AD [7–9]. According to the mitochondrial cascade hypothesis, age-associated loss of mitochondrial function affects the expression and processing of β-amyloid precursor protein (APP), producing Aβ oligomers that accumulate into plaques in AD [9].

*Ginkgo biloba* L. leaf extract (GBE) has a long history of usage in a traditional Chinese medicine and recently been incorporated in numerous commercial herbal products. Several studies indicate that *Ginkgo biloba* extracts provide benefits in the management of brain injury and depression [10–12]. Furthermore, anti-apoptotic and anti-oxidative properties of GBE are well documented, which are mainly due to its ability to scavenge free radicals [13–15]. Mitochondria are major cellular generators of reactive oxygen species (ROS) that cause oxidative damage [3,16]. EGb761, a commercial GBE product mainly composed of *Ginkgo biloba* flavonoids (22–27%) and terpene lactones (5–7%), is widely used as an anti-dementia drug in clinical research [17,18]. Dried *Ginkgo biloba* extract 50 (GBE50) is a novel form of *Ginkgo biloba* extract similar to EGb761 containing *Ginkgo biloba* flavonoids (44%) and terpene lactones (6%) as its active components [19]. One proposed mechanism for the neuroprotective function of EGb761 is preventing activation of mitochondria-mediated apoptotic pathways [20]. However, in recent years, research on GBE50 has predominantly focused on cardiovascular protection and the molecular mechanisms underlying its anti-oxidative activity in AD remain to be established.

In this study, an AD-like rat model was generated via bilateral injection of $A\beta_{25-35}$ into the hippocampal CA1 region. Levels of peroxidation in the hippocampus, particularly in mitochondria, along with oxidative stress and apoptosis induced by peroxidation were subsequently evaluated, with the objective of ascertaining whether GBE50 could inhibit these pathological changes and improve cognitive ability in animals.

## 2. Materials and methods

### 2.1 Materials

GBE50 (Lot No. 111201) with a composition of $\geq 44\%$ *Ginkgo biloba* total flavonoids ($\geq 24\%$ flavonol glycosides, $\geq 20\%$ free flavones), $\geq 6\%$ lactones ($\geq 3.1\%$ *Ginkgo biloba* lactones and $\geq 2.9\%$ bilobalide) and $\leq 5$ parts per million ginkgolic acids was supplied by SPH Xing Ling Sci. & Tech. Pharmaceutical Co., Ltd. (Shanghai, China). The compounds of GBE50 sourced from the above company have been identified via UPLC-Q/TOF-MS analysis in a previous study [21]. The product was prepared as 7.5 g/L and 15 g/L solutions using the solvent 1% (w/v) sodium carboxymethycellulose (CMC). Vitamin E (VE, Lot No.03120202) was purchased from SPH Sine Pharmaceutical Laboratories Co., Ltd., China. The capsule contents (100 mg) were removed, added to 10 mL of 1% CMC solvent and stirred evenly to generate a VE suspension (10 mg/mL). $A\beta_{25-35}$ (No. A4559; Sigma, USA) was dissolved in sterile double-distilled water at a concentration of 5 μg/μL and incubated to induce aggregation at 37°C for 4

days before use. The preparation and aggregation procedures for $A\beta_{25-35}$ were performed according to previously described methods [22,23].

## 2.2 Animals

Male Sprague-Dawley rats ($220 \pm 10$ g) were obtained from the Experimental Animal Center of Shanghai University of Traditional Chinese Medicine (SHUTCM). Rats were housed in groups of six and maintained in a regulated environment ($23°C$, 50% humidity, 12 h light-dark cycle), with *ad libitum* access to food and water. All experiments were approved by the Animal Committee of Shanghai University of Traditional Chinese Medicine (approval number: PZSHUTCM2212260003). The number of animals in the experimental groups was kept to a minimum and all studies were conducted in a manner designed to cause the least harm and suffering to animals.

## 2.3 Hippocampal injection and drug administration

Sixty-five rats were used for study and divided into six experimental groups: Sham (control animals administered vehicle only, sterile water, n = 11), Model ($A\beta_{25-35}$ injection, n = 10), CMC (CMC plus $A\beta_{25-35}$, n = 11), VE (vitamin E plus $A\beta_{25-35}$, n = 11), G1 (75 mg/kg GBE50 plus $A\beta_{25-35}$, n = 11) and G2 (150 mg/ kg GBE50 plus $A\beta_{25-35}$, n = 10). For surgical purposes, rats were anesthetized with 2% isoflurane delivered in $O_2$ at a flow rate of 1 L/min and placed onto a stereotaxic apparatus. Subsequently, 20 μg $A\beta_{25-35}$ was injected into the rat hippocampus [24]. Briefly, 2 μL $A\beta_{25-35}$ aggregates were gradually administered into each CA1 region of bilateral hippocampus with a 10 μL Hamilton syringe. The coordinates for hippocampal injection (3.5 mm posterior to bregma, 2.5 mm lateral to the sagittal suture, 2.9 mm ventral to the dura) were similar to those used by Shen et al. [24]. Sham-operated animals received an equivalent volume of sterile distilled water. Accurate injection was confirmed via histological examination of the brain.

Drugs were administered orally via gastro-esophageal gavage the day after hippocampal injection, once daily for 15 days. The doses of GBE50 (75 and 150 mg/kg/d) and VE (100 mg/kg/d) provided to rats were similar to those established in a previous study [25,26]. The CMC group of rats received 1% CMC solution while the sham and model groups were provided saline solution.

## 2.4 Behavior tests

The Morris water maze (a 150 cm wide and 70 cm high round pool with four styles of white shapes on a black wall and filled to a depth of 50 cm with opaque water at a temperature of $22 \pm 2°C$) was used to examine spatial memory of animals after drug administration. The platform was a cylinder 12 cm in diameter placed in the center of the first quadrant of the pool 30 cm from the wall and submerged 1.5 cm below the water surface. Rats were trained in five daily acquisition sessions (four trials each day with 30 min intervals) to detect the hidden platform. The maximum swimming time was set to 70 s. In cases where the escape latency was < 70 s, rats were allowed to stay on the platform for 10 s. If the platform was not found within 70 s, rats were guided to the platform and stayed on it for 10 s, with the escape latency taken as 70 s. All experiments were recorded using a computerized tracking analyzer system (Shanghai Mobiledatum Ltd, Shanghai, China) [27].

## 2.5 Tissue preparation

Following the behavior tests, all animals were anesthetized with 1% pentobarbital sodium (50 mg/kg). Three rats of each group were transcardially perfused with PBS, followed by 4% formaldehyde in PBS. The hippocampus was post-fixed for 24 h and cryoprotected in phosphate-buffered 30% sucrose. Hippocampal tissue was systematically sliced using a freezing microtome into 20 μm thick sections for histological and immunofluorescence analyses. The remaining rats of each group were decapitated and hippocampi isolated from brain sections on an ice-cold glass plate. Hippocampal tissues were stored at -80˚C for use in biochemical assays.

## 2.6 Preparation of hippocampal mitochondria

The hippocampus was removed to a 1.5 mL EP tube and homogenized in ten volumes of reagents from a Mitochondria Isolation Kit (C3606; Beyotime Biotechnology, Shanghai, China) containing phenylmethyl sulfonyl fluoride on ice. Homogenates were centrifuged (600 ×g) for 5 min at 4˚C, followed by re-centrifugation of the supernatant fraction (11000 ×g) for 10 min at 4˚C. Mitochondrial deposits were resuspended in nine volumes of saline solution, re-homogenized and centrifuged (10000 ×g) for 10 min. The supernatant fraction was used for measurement of $H_2O_2$ and enzymatic activity.

## 2.7 Determination of oxidative stress biomarker levels and enzyme activities

Measurements of $H_2O_2$ production and activities of $H_2O_2$-generating monoamine oxidase (MAO) and antioxidant enzymes, specifically, catalase (CAT), superoxide dismutase (SOD) and glutathione peroxidase (GSH-Px), in mitochondria were performed with the appropriate assay kits ($H_2O_2$, A064; MAO, A034; CAT, A007-1; SOD, A001-3; GSH-Px, A005) from Jiancheng Bioengineering Institute (Nanjing, China).

## 2.8 Western blot

Using the methods developed for preparation of hippocampal mitochondria, proteins of cytoplasm and mitochondria were isolated for evaluation of cytochrome C (Cyto C) and other proteins. Protein concentrations were analyzed with a BCA kit (Weiao Lab, Shanghai, China). Equal amounts of protein (20 μg) were separated via 12% SDS-PAGE and transferred to PVDF membrane (Millipore, Bedford, MA). Next, membranes were blocked with 5% skimmed milk in PBS containing 0.01% Tween-20 for 1 h at room temperature and incubated with the appropriate antibodies in dilution buffer at 4˚C overnight (Table 1). β-Actin was used as the internal

**Table 1. Antibodies used for western blot.**

| Primary AB Name | Producer | Cat. No. | Dilution |
|---|---|---|---|
| Anti-β-actin | CST | #4970 | 1:500 |
| Anti-Caspase-3 | CST | #9662 | 1:500 |
| Anti-Bax | CST | #2772 | 1:500 |
| Anti-Bcl-2 | CST | #2870 | 1:750 |
| Anti-Cyto C | CST | #4272 | 1:500 |
| Anti-CAD | SANTA CRUZ | sc-8342 | 1:200 |
| Secondary AB Name | Producer | Cat. No. | Dilution |
| Gt-α-rb-HRP | ICL | GGHL-15P | 1:5000 |

reference. Films were digitized and the densitometry of each protein band analyzed with the Tanon 4500SF system (Tanon, Shanghai, China).

## 2.9 Immunofluorescence detection

Sections (20 μm) were incubated in PBS for 15 min, treated with 0.1% Triton X-100 in PBS for 15 min at room temperature and blocked in 10% donkey serum for 1 h at 37˚C. Sections were incubated with goat anti-8-OHdG antibody (ab10802, 1:100, Abcam) at 4˚C overnight, followed by Alexa488-conjugated donkey anti-goat (705-547-003, 1:200, Jackson) as the secondary antibody for 1 h at 37˚C. Nuclei were stained with DAPI (1 μg/mL, Sigma) for 10 min in the dark. The coverslips were mounted with mounting medium (Weiao Lab, Shanghai, China). The results of immunostaining were examined using a fluorescence microscope (BX60, Olympus, Japan).

## 2.10 Statistical analysis

All data are expressed as mean ± standard deviation (SD). Statistical analyses were performed using SPSS 16.0 (IBM, USA). One-way ANOVA, followed by Student-Newman–Keuls (S-N-K) multiple comparison test, was applied to assess the statistical significance of results. $P$ values < 0.05 were considered statistically significant.

## 3. Results

### 3.1 Effects of GBE50 on Aβ$_{25-35}$ -induced memory impairment

In the Morris water maze experiment, the escape latency of the model group was prolonged from test day 2 and obviously shorter compared to the sham group on day 5, as shown in Fig 1.

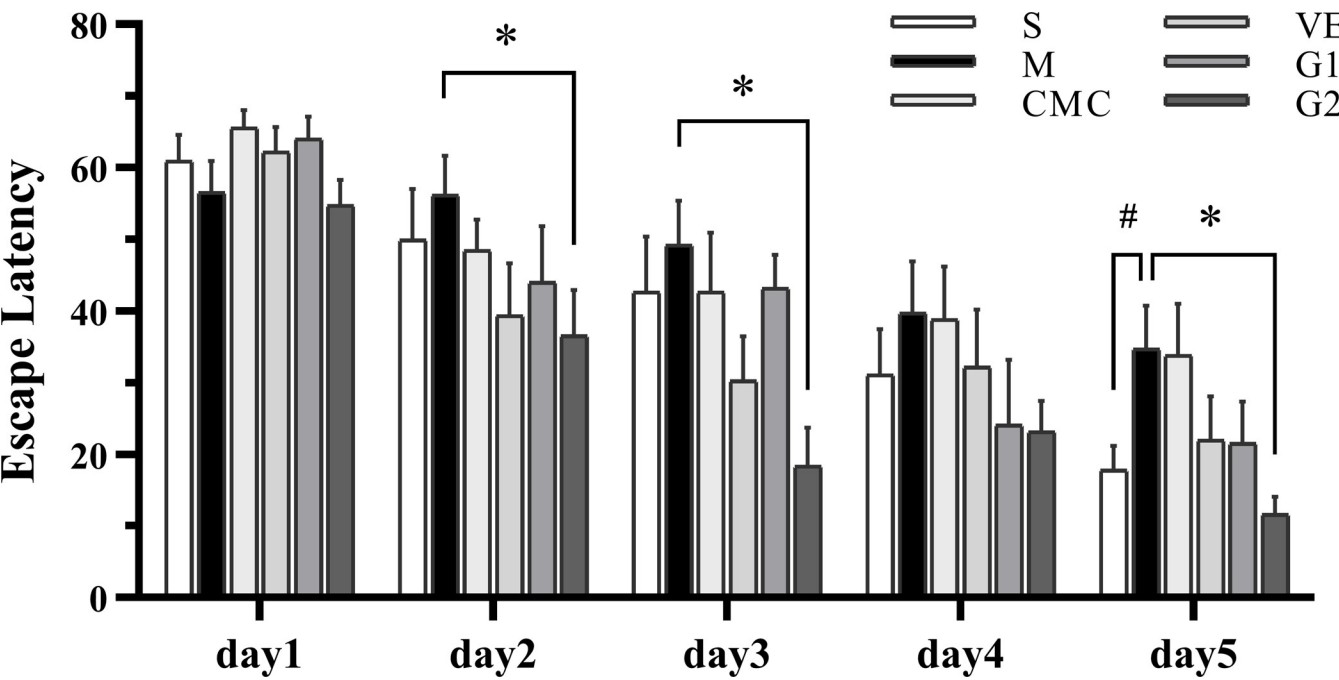

**Fig 1. Escape latency of each group in the behavior test.** Escape latency detected using the Morris water maze was used to examine spatial memory of animals following drug administration. (S: Sham, n = 11. M: Model, n = 10. CMC: Carboxylmethylcellulose, n = 11. VE: Vitamin E, n = 12. G1: 75 mg/kg GBE50, n = 11. G2: 150 mg/kg GBE50, n = 10). Values are expressed as means ± SD, $^\#$ $P$ < 0.05 vs sham, $^*$ $P$ < 0.05 vs model.

No differences were observed between the model and CMC groups. Latencies of both VE and G1 groups were shorter than those of the model group from day 2, but the differences were not statistically significant. The latency of the G2 group was markedly reduced from day 2 in relation to the model group although no significant difference was observed on day 4. Our findings indicate that a high dose of GBE induces a decrease in the escape latency of AD-like rats.

## 3.2 Effects of GBE on $H_2O_2$ formation and enzymatic activity in the hippocampus

$H_2O_2$ is a vital member of the ROS family involved in neurodegeneration. Mitochondria are the primary production and action sites of ROS in cells (Fig 2A). Compared with the sham group, $H_2O_2$ was significantly generated in the CMC group. Elevated $H_2O_2$ was detected in mitochondria of the hippocampus, but this difference was not significant relative to the other groups. Administration of both low and high doses of GBE reduced the $H_2O_2$ level to a non-significant extent compared to the model, CMC and VE groups. In addition, no significant differences in $H_2O_2$ levels were observed among the model, CMC and VE groups (Fig 2B).

The metabolic pathways involving $H_2O_2$ in mitochondria are catalyzed by several enzymes, including monoamine oxidase (MAO) and superoxidase (SOD), which are responsible for generation of $H_2O_2$, as well as glutathione peroxidase (GSH-Px) and catalase (CAT), which catalyze the conversion of $H_2O_2$ to $H_2O$ (Fig 2A). The activity of SOD was markedly diminished in CMC relative to the sham group, and reduced in the model, VE and G1 groups, but not to a significant extent. Elevated activity was detected in G2 compared to the model, CMC and VE groups, but differences were not statistically significant (Fig 2C). MAO activity of the model group was significantly higher compared with the sham group, and activities were

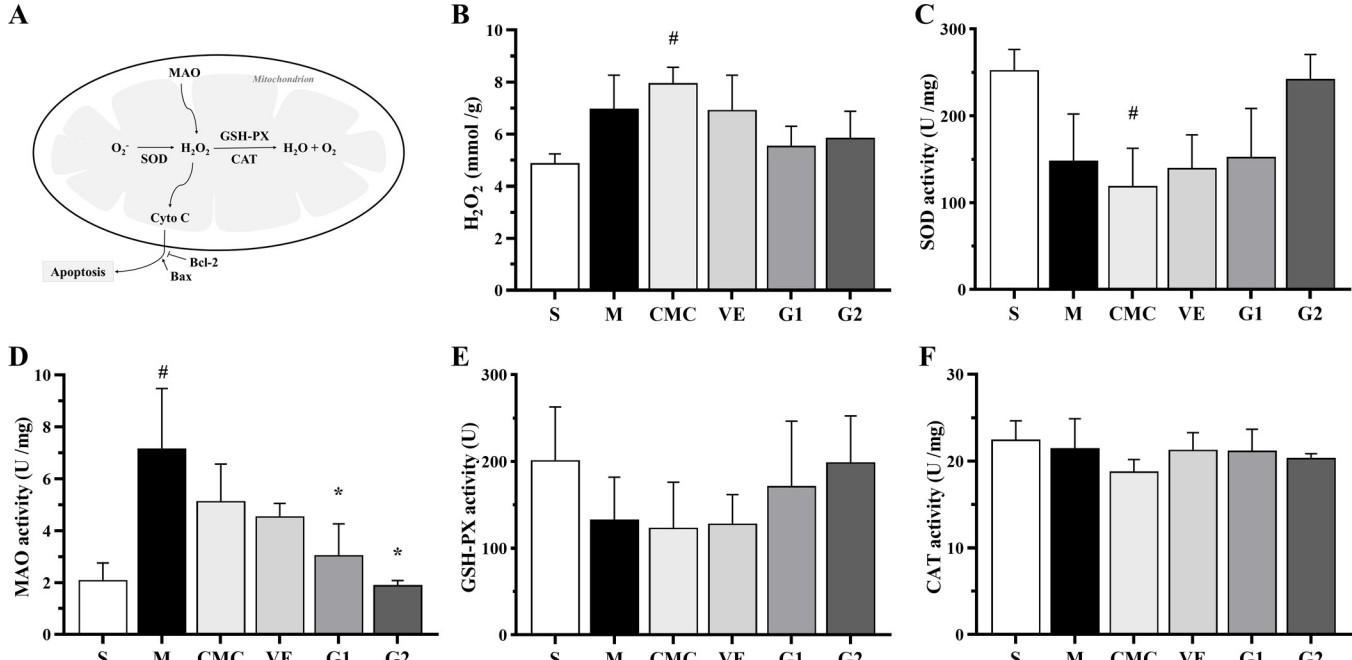

**Fig 2. $H_2O_2$ formation and activities of SOD, MAO, GSH-PX, and CAT in mitochondria of hippocampus.** (A) Mechanism of $H_2O_2$ production and mediation of cytochrome C release from mitochondria into the cytoplasm. (B-F) Detection of $H_2O_2$, SOD, MAO, GSH-PX and CAT activities in all groups. S: Sham, M: Model, CMC: Carboxylmethylcellulose, VE: Vitamin E, G1: 75 mg/kg GBE50, G2: 150 mg/kg GBE50, n = 3. Values are expressed as means ± SD. # $P < 0.05$ vs sham, * $P < 0.05$ vs model.

elevated in CMC, VE and G1 groups, with no statistical differences among the groups. In the G1 and G2 groups, MAO activity was markedly reduced relative to the model group, but no significantly different to that of the VE group (Fig 2D). GSH-Px activity was diminished in the model, CMC and VE groups compared to the sham group. Although its activity was elevated in the G1 group and further in the G2 group, differences were not statistically significant between each of these groups and the model group (Fig 2E). Moreover, differences in activities of SOD and GSH-Px were not statistically significant among the model, CMC and VE groups as well as between VE and GBE groups (Fig 2C and 2E). We observed no significant changes in CAT activity among all the groups (Fig 2F). Our results suggest that GBE inhibits $H_2O_2$ production by suppressing the activity of MAO and concomitantly enhancing those of SOD and GSH-Px.

### 3.3 Effect of GBE on cytochrome C protein expression in hippocampus

Cyto C normally exists in the mitochondrial membrane space and is released into the cytoplasm by the outer membrane at the early stage of apoptosis, promoting the formation of caspase-activated complex apoptotic bodies (Fig 2A) [3]. Western blot analysis showed that relative to the sham group, cytoplasmic Cyto C protein levels were increased significantly and the ratio of cytoplasmic to total intracellular protein was high in the model, CMC, VE and G1 groups, suggesting that high levels of Cyto C are released into the cytoplasm by mitochondria (Fig 3A and 3B). In addition, no significant differences in cytoplasmic Cyto C protein levels were observed among the model, CMC and VE groups. Within the GBE therapy groups, the cytoplasmic protein levels and ratio of cytoplasmic to total intracellular protein were markedly decreased in G2 compared to the model group, indicating that high-dose GBE inhibits release of Cyto C into the cytoplasm from mitochondria.

### 3.4 Effects of GBE on Bcl-2 and Bax protein expression in hippocampus

Bcl-2 and Bax function as anti-apoptotic and pro-apoptotic proteins in the cytoplasm, respectively. Their stabilities are conducive to stabilization of the mitochondrial membrane potential and inhibition of the release of Cyto C by mitochondria (Fig 2A). Compared with the sham group, Bcl-2 expression in the model group was significantly reduced, along with a subnormal level of Bax, leading to an overall decrease in the Bcl-2 to Bax ratio (Fig 3A and 3C). Similarly, low expression of both proteins was maintained in CMC and VE groups. In G1 and G2 treatment groups, levels of the above proteins were higher than those in the model, CMC and VE groups. In particular, Bcl-2 expression was markedly increased compared to the model group and the Bcl-2 to Bax ratio was higher relative to the other groups. Based on the results, we propose that GBE induces Bcl-2 expression that exerts a protective effect on mitochondria and inhibits Cyto C release into the cytoplasm.

### 3.5 Effects of GBE on caspase-3 and CAD protein expression in hippocampus

Caspase-3, a major apoptotic protease in the cytoplasm, either directly participates in DNA fragmentation or indirectly activates endonuclease caspase-activated DNase (CAD) to cleave chromatin and initiate cell apoptosis [3].

Caspase-3, excluding cleaved caspase-3, was evaluated in all the groups. Compared with the sham group, high levels of caspase-3 were observed in the model and CMC groups, but differences were not significant. However, levels of caspse-3 were significantly decreased in the VE, G1 and G2 groups compared to the model group (Fig 3A and 3D). No marked differences in

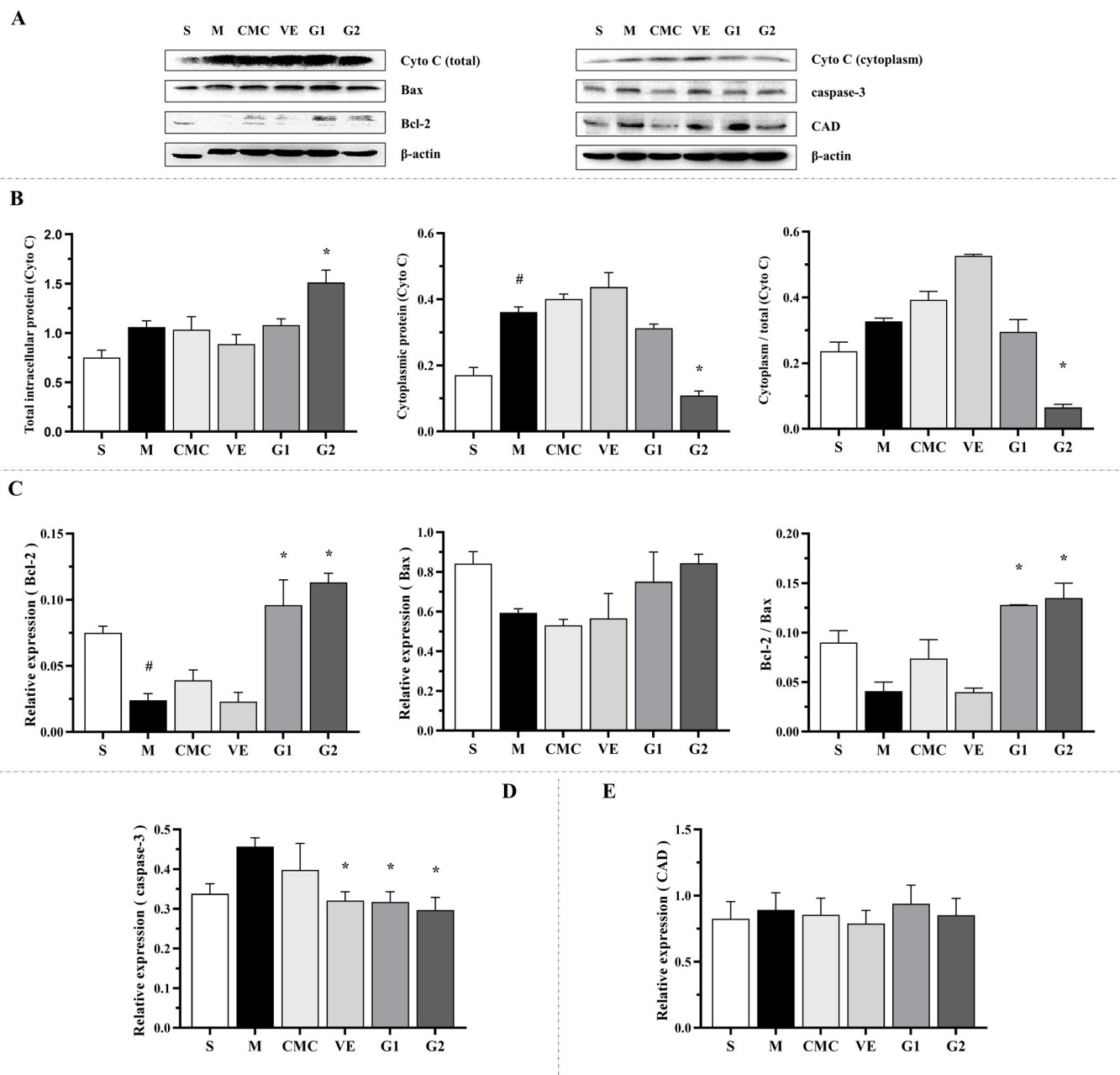

**Fig 3. Relative quantitative expression of intracellular proteins with antioxidant and anti-apoptotic activity in hippocampus.** (A) WB detection of Cyto C (14 kDa), Bax (20 kDa), Bcl-2 (28 kDa), caspase-3 (35 kDa) and CAD (40 kDa) proteins, with β-actin (45 kDa) as the loading control. (B) Relative expression of total intracellular and cytoplasmic proteins of Cyto C, and ratio of cytosolic to total intracellular protein of Cyto C. (C) Relative expression of Bcl-2 and Bax proteins, and ratio of Bcl-2 to Bax. (D) Relative expression of caspase-3 protein. (E) Relative expression of CAD protein. S: Sham, M: Model, CMC: Carboxylmethylcellulose, VE: Vitamin E, G1: 75 mg/kg GBE50, G2: 150 mg/kg GBE50, n = 3. $^{\#}$ $P < 0.05$ vs sham, $^{*}$ $P < 0.05$ vs model.

CAD were observed among the groups (Fig 3A and 3E). Our data suggest that GBE inhibits the expression of caspase-3 while CAD is not influenced by Aβ, VE and GBE.

## 3.6 Effects of GBE on 8-OHdG proteins in CA1 of hippocampus

8-OHdG, one of the most common forms of free radical-induced oxidative lesions, is widely used as a biomarker for oxidative stress and carcinogenesis [28]. Using immunofluorescence

to monitor changes in the 8-OHdG content in CA1 of hippocampus, the degree of DNA damage caused by oxidative stress was evaluated. Compared with the sham group, the 8-OHdG content was significantly increased in the model, CMC and VE groups (Fig 4), and conversely, remarkably decreased in both GBE50 dose groups. The data suggest that 8-OHdG induced by the injection of Aβ is significantly reduced by GBE50 but not vitamin E.

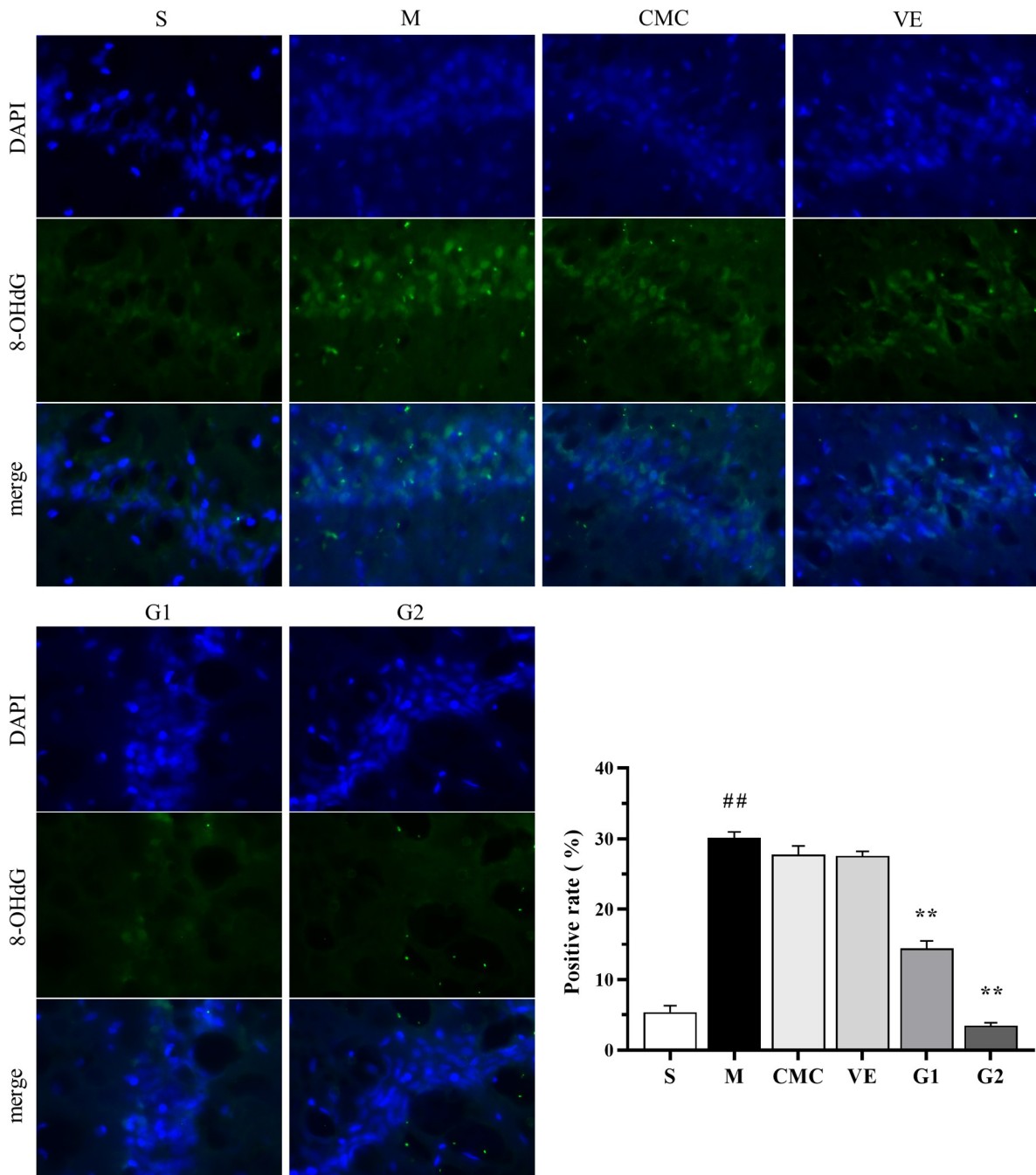

**Fig 4. Immunofluorescence analysis and positive labeling rate of 8-OHdG in hippocampus of each group.** The blue marker represents DAPI and the green marker represents 8-OHdG. (400 x). S: Sham, M: Model, CMC: Carboxylmethylcellulose, VE: Vitamin E, G1: 75 mg/kg GBE50, and G2: 150 mg/kg GBE50, n = 3. [##] $P < 0.01$ vs sham, [**] $P < 0.01$ vs model.

## Discussion

Alzheimer's disease (AD) is a form of senile dementia characterized by a progressive intellectual decline. Accumulation of Aβ plaques is a key pathological mechanism of AD [29], inducing cytotoxicity that damages cells. An AD-like animal model generated via intracranial injection of aggregated Aβ has been widely applied in AD research. Aβ aggregates induce toxicity and nerve tissue damage in animals, resulting in learning and cognitive impairment [22,23]. In preclinical research, *Ginkgo biloba* extracts exerted multiple anti-AD effects. EGb761 inhibits amyloid aggregation through induction of heat shock in aluminum-induced neurotoxicity [30], improves learning and memory in spatial and nonspatial memory tasks, and plays a neuroprotective role in a 5×FAD mouse model [31]. In the current study, escape latencies of animals were prolonged in the model group and reduced in the two dosage groups of GBE50, suggesting that diminished memory ability is induced by $Aβ_{25-35}$ aggregates, which is improved by GBE50.

While Aβ has been shown to induce cytotoxicity, the molecular mechanisms are yet to be fully elucidated. Toxicity is believed to involve the formation of oxygen-free radicals and nitric oxide as well as calcium imbalance [16]. $Aβ_{25-35}$ and $Aβ_{1-40}$ induce a 2-fold increase in intracellular free radicals and 3.5- to 4-fold increase in free calcium [32]. $Aβ_{1-42}$ promotes calcium imbalance in cells mediated by $H_2O_2$ and PAF [33]. $H_2O_2$ in ROS is mainly produced in mitochondria. Mitochondrial breakdown plays a key role in ROS production, in turn, triggering oxidative damage in cells [3,34]. $Aβ_{1-42}$ and $Aβ_{25-35}$ enhance ROS production in mitochondria of rat brain and muscle tissue [35] while $Aβ_{1-40}$ promotes $H_2O_2$ in mitochondria of rat brain [36]. In this study, $H_2O_2$ was induced by $Aβ_{25-35}$ in hippocampal mitochondria of AD rats, which was attenuated by GBE50, supporting the theory that *Ginkgo biloba* extracts exert beneficial effects on scavenging free radicals [37]. For example, *Ginkgo biloba* extracts regulate the oxidative phosphorylation system of the respiratory chain in mitochondria to reduce production of ROS and repair mitochondrial dysfunction induced by Aβ [15]. EGb761 inhibits gentamicin-induced ototoxicity by reducing the production of ROS and NO in isolated rat cochlear hair cells and inhibiting apoptosis of cochlear hair cells *in vivo* [38]. In addition, EGb761 inhibits zinc-induced tau phosphorylation at Ser262 through its anti-oxidative activity [39].

Aβ promotes $H_2O_2$ production, mainly through promoting an imbalance between ROS and antioxidant levels in mitochondria. The $H_2O_2$ level in mitochondria depends on the activities of $H_2O_2$-producing (Mn-SOD in the matrix and MAO in the outer membrane of mitochondria) and $H_2O_2$-consuming enzymes (CAT and GSH-Px) [3]. Continuous lateral ventricle injection of $Aβ_{1-40}$ has been shown to enhance the activities of Cu-Zn SOD in the cytoplasm of rat neocortex, promote MAO-B activity and reduce the activities of CAT and GSH-Px in mitochondria to a significant extent [40]. In our experiments, upon injection of $Aβ_{25-35}$ into the CA1 region of the hippocampus, MAO activity was enhanced and GSH-Px and SOD activities diminished while CAT activity was not affected. The observed changes of the activities of SOD and CAT were inconsistent with earlier findings, which could be attributed to different Aβ preparations or models.

In our experiments, GBE50 suppressed MAO activity and enhanced the activities of SOD and GSH-Px, while exerting no significant effects on CAT activity. The data clearly indicate that GBE50 affects the activities of $H_2O_2$-producing and -consuming enzymes. GBE50 could inhibit the production of $H_2O_2$ by suppressing the activity of MAO and enhance the activities of SOD and GSH-Px to accelerate $H_2O_2$ generation and metabolism into $H_2O$ in a timely manner, therefore eliminating ROS and inducing resistance of mitochondria to Aβ damage. Numerous studies have focused on regulation of ROS-related enzyme activities by *Ginkgo biloba* leaf extract. EGb761 stabilizes the redox state of cells by upregulating protein levels and

antioxidant enzyme activity [41], increases the activities of SOD and CAT in hippocampus of rats [42], enhances total superoxide dismutase (T-SOD), CAT, and GSH-Px activities in Neuro 2A cells overexpressing APPsw [43], and promotes the activities of glutathione reductase and gamma-glutathione synthase, two key enzymes involved in glutathione (GSH) reduction and synthesis [44]. However, EGb761 could reduce protein GSH and activity of GSH-Px in rats exposed to an intermittent hypoxia environment consecutively for 21 days [45]. These differences may be caused by different animal models and other factors. In this study, GBE50 was expected to regulate the activity of $H_2O_2$-producing enzymes in mitochondria and enhance that of $H_2O_2$-consuming enzymes to exert anti-oxidative effects.

Expression of Bcl-2 in hippocampal tissues of the model group was significantly diminished, along with a low Bcl-2/Bax ratio and marked increase in expression of Cyto C in cytoplasm. The results suggest that Aβ induces $H_2O_2$ production in mitochondria of hippocampal tissue, reduces expression of Bcl-2 protein, and accelerates the release of Cyto C from mitochondria, causing damage to tissue cells. Earlier studies have shown that Aβ induces cellular mitochondrial dysfunction and stimulates release of Cyto C into the cytoplasm by mitochondria. Under physiological conditions, Cyto C exists in the mitochondrial membrane gap and is released into the cytoplasm by mitochondrial outer membrane at the early stage of apoptosis, promoting the formation of caspase-activated complex apoptotic bodies [3]. Aβ induces activation of the caspase cascade and affects cell death pathways. For example, *in vitro*, Aβ$_{25-35}$ and Aβ$_{1-40}$ are reported to activate caspase-8 and caspase-3, interfere with nuclear and mitochondrial DNA integrity, and induce apoptosis of cerebral vascular endothelial cells in mouse and bovine models [46]. Aβ$_{25-35}$ induces cell body contraction of cerebellar granulosa cells, neurite retraction, changes in mitochondrial activity, and enhancement of caspase-3 activity [47]. Aβ isolated from early-onset familial AD patients inhibits the proliferation and differentiation of cultured human and rodent neural progenitor cells by promoting apoptosis [48]. In an *in vivo* study by Kaminsky et al. [3] involving injection of Aβ$_{25-35}$ or Aβ$_{1-40}$ into the lateral ventricle of rats consecutively for 14 days, mitochondria of the cerebral cortex released Cyto C to the cytoplasm, along with a 2- to 3-fold increase in activities of caspase-3 and caspase-9. Furthermore, treatment with EGb761 led to a significant decrease in cell viability and apoptosis in response to incubation with Aβ$_{1-42}$ oligomer [49]. The effects of GBE50 on expression patterns of the above proteins were explored in the current study. Expression of Bcl-2 and Bcl-2/Bax ratio in both GBE50 dose groups were markedly increased while caspase-3 (excluding cleaved caspase-3) and Cyto C levels were significantly diminished in the GBE50 high-dose group. Based on these results, we propose that GBE50 inhibits Aβ-mediated induction of $H_2O_2$ in hippocampal mitochondria and enhances Bcl-2 protein expression to inhibit release of Cyto C to the cytoplasm, which protects hippocampal nerve tissue from oxidative stress injury. The anti-apoptotic effect of EGb761 may be achieved through synergistic multiple intracellular signaling pathways, including maintaining integrity of the mitochondrial membrane, preventing mitochondria from releasing Cyto C to inhibit formation of the apoptotic complex and caspase apoptotic proteases, enhancing transcription of anti-apoptotic Bcl-2-like protein, impairing the transcription of pro-apoptotic caspase-12, and inhibiting the main apoptotic effector protease, caspase-3, to prevent execution of apoptosis and formation of nuclear DNA fragments [50,51]. For example, EGb761 regulates expression of the apoptosis-related proteins Bcl-2 and Bax to inhibit $H_2O_2$-induced neuronal death [52,53]. Furthermore, EGb761 inhibits the release of Cyto C from mitochondria of cardiomyocytes induced by hypoxia and reoxygenation, reduces caspase-3 activity and inhibits DNA fragmentation to suppress myocardial cell apoptosis [54]. EGb761 inhibits ROS generation, activates SOD activity, maintains homeostasis of Bcl-2 family proteins and stabilizes mitochondrial membrane potential to inhibit release of Cyto C by mitochondria, which protects against oxidative stress in human umbilical vein

endothelial cells [55]. The mechanisms underlying GBE50-mediated inhibition of Aβ damage to hippocampus of AD-like animals uncovered in this study appear consistent with data from the earlier research.

Caspase-activated DNase (CAD) is an endogenous enzyme that mediates DNA nucleosome degeneration [56]. Caspase-3 can either participate in DNA fragmentation directly or indirectly activate CAD and cleave chromatin to induce apoptosis [23]. *In vivo* experiments have shown that following continuous injection of $A\beta_{25-35}$ into rat lateral ventricles, CAD activity in the cerebral cortex, cerebellum and hippocampus is enhanced significantly, accompanied by nuclear DNA fragmentation [3]. In this study, no significant changes in the expression of CAD were observed and cleaved caspase-3 was not detected in all groups, suggesting that Aβ does not affect expression of CAD and activate caspase-3. These findings were inconsistent with the above studies, which could be attributable to different animal models. The results additionally suggest that the effects of GBE50 on DNA injury induced by oxidative stress in hippocampal mitochondria are not directly dependent on activation of caspase-3 and stimulation of CAD expression.

8-OHdG is a major form of free radical-induced oxidative lesions. A number of studies have shown that *Ginkgo biloba* extract affects 8-OHdG levels in liver, serum and brain. For example, pretreatment with n-acetylcysteine (NAC) and EGb761 has been shown to reduce the formation of 8-OHdG and lipid peroxidation in liver tissue of rats [57]. Moreover, EGb761 could suppress the 8-OHdG level in serum and hippocampus of rats induced by intermittent hypoxia [45]. In the current study, GBE50 suppressed the formation of 8-OHdG in the hippocampus of AD rats, which inhibited the effect of Aβ-induced oxidative stress on DNA injury.

The constituents in GBE50 were analyzed by UHPLC-Q-Exactive Orbitrap HRMS in a current study. Altogether 38 compounds were analyzed in GBE50, including 23 flavonoids, 5 biflavonoids, 4 catechins, 5 terpene lactones, and 1 organic acid [58]. The active constituents of GBE include flavonoids (e.g., quercetin, kaempferol, and isorhamnetin), biflavones (sciadopitysin and ginkgetin), terpene trilactones (ginkgolides and bilobalide), and ginkgolic acids (alkylphenols) [59]. A variety of studies have shown that several active constituents of GBE have significant antioxidant effects. In GBE, ginkgo flavonoids, proanthocyanidins, and organic acids have a large number of reduced hydroxyl functional groups, which can play an antioxidant role by scavenging oxygen free radicals and regulating the activity of superoxide dismutase and catalase [60]. Kaempferol, one of the most important constituents of Ginkgo biloba, reduces ROS generation by scavenging free radicals, upregulates Bcl-2 and glutathione (GSH) to protect neuronal cells from oxidative injury [61], and inhibits mitochondrial membrane transition (mPTP) opening and suppresses the release of Cyto C via glycogen synthase kinase-3β inhibition [62]. Besides, kaempferol can inhibit Bax and caspase-3 to exert anti-apoptotic effects [63]. Ginkgetin and bilobalide decrease levels of intracellular ROS, maintain mitochondrial membrane potential, and inhibit cell apoptosis via caspase-3 and Bcl2/Bax pathways to exert antioxidant effects in the mouse model of Parkinson's disease [64,65]. Isorhamnetin reduces activation of the extrinsic apoptotic pathway by decreasing caspase-3 and caspase-8 in the cell model of ischemia-induced cerebral vascular degeneration [66]. In addition, Ginkgolide B improves antioxidant defense system (SOD, GSH and CAT) in hippocampal tissue of rats treated with hypoxia exposure for six days [67].

Although the various chemical components of GBE are integrated, complementary and synergistic interactions exist between several members. Zhang et al. reported that any two members of four typical components of GBE can exhibit apparent synergistic antioxidant effects, and the ginkgo flavone: procyanidins (1: 9) showed the best synergism in scavenging DPPH (2,2-Di(4-tert-octylphenyl)-1-picrylhydrazyl) and ABTS (2'-Azinobis-(3-ethylbenzthiazoline-6-sulphonate) radicals [60].

However, for GBE50, few studies focus on the antioxidant activity in the neurodegenerative disease. Lu et al. observed that pre-treatment of GBE50 dose-dependently significantly increased myocardium SOD, CAT, GSH-Px and GST activities to exert antioxidant effects in ischemia reperfusion rats [19]. In our study, antioxidant and anti-apoptotic activities of GBE50 were observed in rat model of AD. According to these reported studies, we suppose that the antioxidant effect observed in this study would rely on the complementary and synergistic interactions existed among active compounds of GBE50.

In conclusion, GBE50 inhibits hippocampal mitochondrial oxidative stress induced by Aβ to exert protective effects on nerve tissue and improve the learning and memory abilities of AD-like rats. The underlying mechanisms potentially involve regulation of the activities of enzymes that play a role in $H_2O_2$ metabolism (MAO, SOD and GSH-Px) to reduce $H_2O_2$ generation in mitochondria, modulation of Bcl-2 and Bax proteins to inhibit release of Cyto C from the mitochondria to cytoplasm, and reduction of 8-OHdG production to avoid neuronal injury from oxidative stress. Our results clearly support a protective role of GBE50 against oxidative stress. As a promising natural agent for AD [68], the therapeutic effects of *Ginkgo biloba* extract may be achieved through synergism of multiple intracellular signaling pathways [69], synergistic interactions existed among active compounds of GBE and promotion of hippocampal neurogenesis in the context of brain aging [70]. Therefore, the mechanism uncovered in this study may be only one of the contributory aspects to GBE50-mediated resistance to tissue oxidative stress and cell damage, highlighting the necessity for further research to establish the mechanistic network.

## Supporting information

**S1 Dataset.**
(RAR)

## Author Contributions

**Conceptualization:** Chenyi Xia, Zhixiong Zhang.

**Data curation:** Xianwen Dong, Meifang Jiang.

**Formal analysis:** Yan Zhao.

**Funding acquisition:** Chenyi Xia.

**Methodology:** Mingmei Zhou, Xianwen Dong.

**Project administration:** Chenyi Xia, Xianwen Dong, Yan Zhao, Guoqin Zhu, Zhixiong Zhang.

**Resources:** Zhixiong Zhang.

**Supervision:** Guoqin Zhu, Zhixiong Zhang.

**Validation:** Chenyi Xia, Meifang Jiang, Guoqin Zhu.

**Writing – original draft:** Chenyi Xia, Mingmei Zhou.

**Writing – review & editing:** Chenyi Xia, Mingmei Zhou, Guoqin Zhu.

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
