## [Decision Letter · Decision Letter 0]

20 Oct 2023

PONE-D-23-22230Ginkgo biloba extract inhibits hippocampal neuronal injury caused by mitochondrial oxidative stress in Alzheimer's disease ratPLOS ONE

Dear Dr. Xia,

Thank you for submitting your manuscript to PLOS ONE. After careful consideration, we feel that it has merit but does not fully meet PLOS ONE’s publication criteria as it currently stands. Therefore, we invite you to submit a revised version of the manuscript that addresses the points raised during the review process.

We look forward to receiving your revised manuscript.

Kind regards,

Tosin Abiola Olasehinde

Academic Editor

PLOS ONE

Journal Requirements:

3. Please remove your figures from within your manuscript file, leaving only the individual TIFF/EPS image files, uploaded separately. These will be automatically included in the reviewers’ PDF.

Additional Editor Comments:

Authors should revise the manuscript appropriately and address all comments indicate by the reviewers

Abstract

The concentration of the ginkgo biloba extract 50 was not specified

Administration of GBE50 can reverse the activity of these three enzymes and reduce the level of H2O2 – this is not clear. The study has been done already, why did authors indicate it can reverse. The use of grammar in the abstract is not appropriate

Methods

Section 2.2 animals - Male SD rats (220 ± 10 g) – what kind of animals were used? SD is not defined.

Line 94 – what is concentration of Aβ25-35 used?

How was Aβ25-35 prepared and aggregated? For how long was it prepared before administration. Was it a once off administration or continuous?

How did the authors confirm that the rats had Alzheimer’s disease before administering the drug? Was a confirmatory test done on all the rats before GBK was administered?

Alzheimer’s disease is a progressive disease and may be developed overtime especially. Why did authors choose to administer GBK a day after Aβ25-35 injection?

What is the justification for the dose (n at dose of 75 or 150 mg/kg) and 15 days treatment

Line 103 – authors indicated Behaviour Tests but only one test (morris water maze) was done.Why did authors conduct only behavioural test. This cannot reveal a true picture behavioural function of the animal.

Results

Line 172 – 174 – the statistical analysis in figure 2b and footnote contradicts the arguments/explanation of the results in the result section (line 172-174(. Authors should check their statistical analysis to corroborate this explanation.

Similarly in Line 180 – 183 – MAO activity was weaker in CMC and VE, but the comparism cannot be seen in Figure 2D. The statistical analysis does not reveal this effect in Figure 2D

Authors should check the statistical analysis in all the tables and comparisms of the groups and revise appropriately

Line 185 – each them should be corrected

Line 183 – 186 and Figure 2c and e are confusing, diminishing of the enzyme but

Authors should revise the issues around the interpretation of the statistical analysis

Group comparism should be done

The report of the results is not specific,

Line 241 – authors should clarify the phrase “caused by aging Aβ.” The methods did not reveal anything related to aging being induced in the animals

Line 265 – 266 - It had been 266 confirmed that Aβ can change the activity of H2O2-related enzymes in cells in vitro – who confirmed this

Line 270 – 271 should be revised

Line 357 – 358 is confusing - include adjusting the ROS 358 metabolism enzyme activity to reduce H2O2 generation in the mitochondria

There are lots of grammatical errors in the manuscript. All sections should be checked for proper use of English

Reviewer's comments 1

I: ginkgo biloba capital letter for Ginkgo and italic

ad libitum italic

in vitro italic

in vivo italic

II: Reference for GBE, Aβ 25-35 , Vitamin E, and CMC doses.

III: Writing the references does not match the Plos one journal method.

Reviewer 2

It is necessary to investigate the proteins of the internal pathway of apoptosis.

The following proteins should be measured.

caspase 3

cleaved caspase 3

caspase 9

cleaved caspase 9

Measurement of amyloid beta peptide is necessary.

The quality of the images is low.

Reviewer 3

1. In the introduction section, the authors need to mention the impact of mitochondria oxidative stress in the pathogenesis and progression of Alzheimer’s disease

2. In the methodology section, the subheading “Determination of H2O2 and enzyme activities” should be changed to “determination of oxidative stress biomarkers and enzymes activities”

3. In the discussion section, authors need to thoroughly discuss the impact of their findings on the role of mitochondrial oxidative stress in Alzheimer’s disease. The discussion should provide a link of all the results for easy understanding on the therapeutic effect of Ginkgo biloba extract on Alzheimer’s disease.

4. What are the bioactive components of the extract? It is imperative that the authors identify the bioactive compounds of the extracts which may be responsible for its therapeutic effect.

Reviewers' comments:

Reviewer's Responses to Questions

**Comments to the Author**

1. Is the manuscript technically sound, and do the data support the conclusions?

Reviewer #1: Yes

Reviewer #2: Yes

Reviewer #3: Partly

2. Has the statistical analysis been performed appropriately and rigorously? 

Reviewer #1: Yes

Reviewer #2: Yes

Reviewer #3: Yes

3. Have the authors made all data underlying the findings in their manuscript fully available?

Reviewer #1: Yes

Reviewer #2: Yes

Reviewer #3: Yes

4. Is the manuscript presented in an intelligible fashion and written in standard English?

Reviewer #1: Yes

Reviewer #2: Yes

Reviewer #3: Yes

5. Review Comments to the Author

Reviewer #1: Minor Revision

I:

ginkgo biloba capital letter for Ginkgo and italic

ad libitum italic

in vitro italic

in vivo italic

II: Reference for GBE, Aβ 25-35 , Vitamin E, and CMC doses.

III: Writing the references does not match the Plos one journal method.

Reviewer #2: It is necessary to investigate the proteins of the internal pathway of apoptosis.

The following proteins should be measured.

caspase 3

cleaved caspase 3

caspase 9

cleaved caspase 9

Measurement of amyloid beta peptide is necessary.

The quality of the images is low.

Reviewer #3: 1. In the introduction section, the authors need to mention the impact of mitochondria oxidative stress in the pathogenesis and progression of Alzheimer’s disease

2. In the methodology section, the subheading “Determination of H2O2 and enzyme activities” should be changed to “determination of oxidative stress biomarkers and enzymes activities”

3. In the discussion section, authors need to thoroughly discuss the impact of their findings on the role of mitochondrial oxidative stress in Alzheimer’s disease. The discussion should provide a link of all the results for easy understanding on the therapeutic effect of Ginkgo biloba extract on Alzheimer’s disease.

4. What are the bioactive components of the extract? It is imperative that the authors identify the bioactive compounds of the extracts which may be responsible for its therapeutic effect.

6. PLOS authors have the option to publish the peer review history of their article (what does this mean?). If published, this will include your full peer review and any attached files.

Reviewer #1: No

Reviewer #2: **Yes: **Mohammad Amin Dehghani

Reviewer #3: No

---

## [Author Response · Author response to Decision Letter 0]

6 Dec 2023

Dear Dr. Olasehinde and reviewers,

First I would like to thank you for giving us the opportunity to submit a revised draft of the manuscript “Ginkgo biloba extract inhibits hippocampal neuronal injury caused by mitochondrial oxidative stress in Alzheimer's disease rat” for publication in the Journal of PLOS ONE. We sincerely appreciate the time and effort that you and three reviewers dedicated to provide the valuable comments on our manuscript. All the insightful comments were considered carefully and the manuscript was revised with tracked changes as necessary based on the suggestions made by reviewers. Attached please find the point-by-point response to the reviewers’ comments and concerns. Should you have any questions or concerns, please contact me. 

Sincerely,

Chenyi Xia

---

## [Decision Letter · Decision Letter 1]

14 Feb 2024

PONE-D-23-22230R1Ginkgo biloba extract inhibits hippocampal neuronal injury caused by mitochondrial oxidative stress in Alzheimer's disease ratPLOS ONE

Dear Dr. Xia,

Thank you for submitting your manuscript to PLOS ONE. After careful consideration, we feel that it has merit but does not fully meet PLOS ONE’s publication criteria as it currently stands. Therefore, we invite you to submit a revised version of the manuscript that addresses the points raised during the review process.

**Authors needs to provide detailed information on the characterization and identification of bioactive principles in the GBE extract which contributed to the activity described by the authors. **

We look forward to receiving your revised manuscript.

Kind regards,

Tosin Abiola Olasehinde

Academic Editor

PLOS ONE

**Additional Editor Comments:**

The authors are yet to provide the information on the characterization and identification of the bioactive constituents of GBE. The journal policy requires that information be provided on identification and characterization of extracts.

Please note that this manuscript describes the use of traditional medicinal compounds or extracts that may not be sufficiently chemically characterized.

Reviewers' comments:

Reviewer's Responses to Questions

**Comments to the Author**

1. If the authors have adequately addressed your comments raised in a previous round of review and you feel that this manuscript is now acceptable for publication, you may indicate that here to bypass the “Comments to the Author” section, enter your conflict of interest statement in the “Confidential to Editor” section, and submit your "Accept" recommendation.

Reviewer #1: All comments have been addressed

Reviewer #2: (No Response)

2. Is the manuscript technically sound, and do the data support the conclusions?

Reviewer #1: Yes

Reviewer #2: (No Response)

3. Has the statistical analysis been performed appropriately and rigorously? 

Reviewer #1: Yes

Reviewer #2: (No Response)

4. Have the authors made all data underlying the findings in their manuscript fully available?

Reviewer #1: Yes

Reviewer #2: (No Response)

5. Is the manuscript presented in an intelligible fashion and written in standard English?

Reviewer #1: Yes

Reviewer #2: (No Response)

6. Review Comments to the Author

Reviewer #1: (No Response)

Reviewer #2: (No Response)

7. PLOS authors have the option to publish the peer review history of their article (what does this mean?). If published, this will include your full peer review and any attached files.

Reviewer #1: No

Reviewer #2: **Yes: **Mohammad Amin Dehghani

---

## [Author Response · Author response to Decision Letter 1]

21 Feb 2024

Thanks a lot for your great advices. The compounds of GBE50 had been identified by UPLC-Q/TOF-MS analysis in previous study [1], and this drug was from the same company as the one we bought. We also upload a supporting information of the certificate of analysis for GBE50 provided by the company.

[1] Ke J, Li MT, Huo YJ, Cheng YQ, Guo SF, Wu Y, et al. The Synergistic Effect of Ginkgo biloba Extract 50 and Aspirin Against Platelet Aggregation. Drug Des Dev Ther. 2021;15:3543-60.

---

## [Editor Report · Decision Letter 2]

12 Mar 2024

PONE-D-23-22230R2Ginkgo biloba extract inhibits hippocampal neuronal injury caused by mitochondrial oxidative stress in Alzheimer's disease ratPLOS ONE

Dear Dr. Xia,

Thank you for submitting your manuscript to PLOS ONE. After careful consideration, we feel that it has merit but does not fully meet PLOS ONE’s publication criteria as it currently stands. Therefore, we invite you to submit a revised version of the manuscript that addresses the points raised during the review process.

**Authors should extensively revise the manuscript as indicated in the reviewer's comments. The manuscript requires extensive English editing. The result section should also be thoroughly revised appropriately**==============================

We look forward to receiving your revised manuscript.

Kind regards,

Tosin Abiola Olasehinde

Academic Editor

PLOS ONE

Additional Editor Comments:

Authors did not carry out thorough revision of the manuscript

Abstract

- Line 22 and 23 – two concentrations should be removed

- Line 28 – Oxiditive stress should be oxidative stress

- Line 37 suggested should be suggest

Introduction

- Line 61 - EGb761, as a commercial product of GBE, - what is the difference between EGb761 and GBE. Does processing of the commercial product affect the levels of the bioactive constitutents.

- Line 64 – that similar should be that is similar.

- Line 64 and 65 – what does authors mean and its effect component contains……this is not appropriate and should be revised

- Line 65 – authors only mentioned the class of compounds – flavonoids and lactones present in GBE-50. What are the specific constituents present in GBE50 which has been published as indicated in reference [21]. This should be stated appropriately

- Line 70 – the phrase - A rat model of AD-like is not appropriate. Should be corrected

- Line 115 should be Vit E was administered at 100 mg/kg according to previous study

Results

- The result section should be completely revised. The level of significance should be indicated.

- The presentation of the results which was revised is very poor grammatically. Authors need to employ the service of an English editor to edit the entire manuscript

- Line 192 and 195, no statistical difference at what level?

- Line 199 – comparing should be compared

- No sequential order for the presentation of the results. Presentation of results from Figure 2D must come be after 2B and 2C

- Line 198-199 – MAO activity was increase in which group compared to Sham?

- Line 198 – 200 should be revised. It is confusing considering elevated in the CMC and, VE and G1 groups without statistical difference

- Line 203 – comparing should be compared

- Line 210 – there were on statistical difference – this grammatical expression is not correct

- Line 219 – authors should clarify (Fig 2 A) (3) – cyto C protein expression

- Line 231 – authors should clarify (Fig 2 A) as BcL

- Line 332 – suggested should be suggest

- Line 349 - Furthermore, the result has had indicated that EGb761 – this is not correct

- Line 394 – 395 – should be corrected - and the mechanisms include may be involve the adjusting the activity of H2O2 metabolism enzymes

- Grammatical error in line 396

- Authors should note that the constituents of GBE50 majorly contributed either individually or synergistically to the activities reported in this study. The role of the constituents in the biochemical mechanism of GBE is missing in the discussion. Furthermore, the antioxidant activity of GBE50 which has been reported may also play an important role in the observed activity reported, but all these arguments are missing.

- The manuscript requires extensive English editing

---

## [Author Response · Author response to Decision Letter 2]

26 Apr 2024

Journal: PLOS NOE

Title: Ginkgo biloba extract inhibits hippocampal neuronal injury caused by mitochondrial oxidative stress in a rat model of Alzheimer's disease

PONE-D-23-22230R2

Response to Reviewers

Dear Dr. Olasehinde and reviewers,

First I would like to thank you for giving us the opportunity to submit a revised draft of the manuscript “Ginkgo biloba extract inhibits hippocampal neuronal injury caused by mitochondrial oxidative stress in a rat model of Alzheimer's disease” for publication in the Journal of PLOS ONE. We sincerely appreciate the time and effort that you and three reviewers dedicated to provide the valuable comments on our manuscript. All the insightful comments were considered carefully and the manuscript was revised thoroughly with tracked changes as necessary based on editor’s suggestions. Attached please find the point-by-point response to the editor’s comments and concerns. Besides, we edited three figures properly to match with the results. Should you have any questions or concerns, please contact me. 

Sincerely,

Chenyi Xia

Editor’s Comments to the Authors:

Introduction - Line 61 - EGb761, as a commercial product of GBE, - what is the difference between EGb761 and GBE. Does processing of the commercial product affect the levels of the bioactive constituents.

Author response: Thanks a lot for your great advices. GBE50, a standardized product of Ginkgo biloba extract (GBE) that is the equivalent to the standardized German product as EGb761, has been approved for clinically use by the China Food and Drug Administration (approval number: Z20000049) [1]. GBE50 contains Ginkgo biloba flavonoids (44%) and terpene lactones (6%), EGb761 includes Ginkgo biloba flavonoids (22-27%) and terpene lactones (5-7%) which is mentioned in Introduction of the manuscript. Distinguishing technique in production process contributes to different levels of the bioactive constituents in GBE50 and EGb761, but how the processing of the commercial product affects the difference is commercial secret we don’t know. A recent study has stated that the biological activities of GBE produced by different production processes are different [2]. In addition, we have uploaded a supporting information of the certificate of analysis for GBE50 provided by the company. 

[1] Ke J, Li MT, Huo YJ, Cheng YQ, Guo SF, Wu Y, et al. The Synergistic Effect of Ginkgo biloba Extract 50 and Aspirin Against Platelet Aggregation. Drug Des Dev Ther. 2021;15:3543-60.

[2] Zhang L, Zhu C, Liu X, Su E, Cao F, Zhao L. Study on Synergistic Antioxidant Effect of Typical Functional Components of Hydroethanolic Leaf Extract from Ginkgo Biloba In Vitro. Molecules. 2022; 27(2):439. doi: 10.3390/molecules27020439.

Introduction - Line 64 and 65 – what does authors mean and its effect component contains……this is not appropriate and should be revised.

Author response: Thank you. The description of “its effect components contains……” has been edited properly. These words express that the bioactive components have different levels in EGb761 and GBE50. GBE50 contains Ginkgo biloba flavonoids (44%) and terpene lactones (6%), but EGb761 includes Ginkgo biloba flavonoids (22-27%) and terpene lactones (5-7%) which is mentioned in Introduction of the manuscript. They are two different commercial product of Ginkgo biloba extract.

Introduction - Line 65 – authors only mentioned the class of compounds – flavonoids and lactones present in GBE-50. What are the specific constituents present in GBE50 which has been published as indicated in reference [21]. This should be stated appropriately

Author response: Thank you. The specific constituents present in GBE50 has been stated in Materials of the manuscript. GBE50 with a composition of ≥ 44% Ginkgo biloba total flavonoids (≥ 24% flavonol glycosides, ≥ 20% free flavones), ≥ 6% lactones (≥ 3.1% Ginkgo biloba lactones and ≥ 2.9% bilobalide) and ≤ 5 ppm ginkgolic acids has been identified via UPLC-Q/TOF-MS analysis.

Results - No sequential order for the presentation of the results. Presentation of results from Figure 2D must come be after 2B and 2C.

Author response: Thank you very much. The presentation of results has been adjusted appropriately.

- Authors should note that the constituents of GBE50 majorly contributed either individually or synergistically to the activities reported in this study. The role of the constituents in the biochemical mechanism of GBE is missing in the discussion. Furthermore, the antioxidant activity of GBE50 which has been reported may also play an important role in the observed activity reported, but all these arguments are missing.

Author response: Thanks for the comments. The arguments about antioxidant effects of essential constituents of GBE and antioxidant activity of GBE50 have been edited in the discussion of manuscript.

- The manuscript requires extensive English editing

Author response: Thanks a lot. The manuscript has been edited by a professional native speaker. The detail please refer to the manuscript (with track changes in word).

In addition, we edited three figures (Fig 3, Fig 4 and Fig5) into a new figure (Fig 3 in the manuscript) properly to match with the results. Fig 1 and Fig 2 were kept the original, and Fig 6 of the previous edition of manuscript was switch to Fig 4 in the current manuscript.

---

## [Editor Report · Decision Letter 3]

19 Jun 2024

PONE-D-23-22230R3Ginkgo biloba extract inhibits hippocampal neuronal injury caused by mitochondrial oxidative stress in a rat model of Alzheimer's diseasePLOS ONE

Dear Dr. Xia,

Thank you for submitting your manuscript to PLOS ONE. After careful consideration, we feel that it has merit but does not fully meet PLOS ONE’s publication criteria as it currently stands. Therefore, we invite you to submit a revised version of the manuscript that addresses the points raised during the review process.

Comment from Editor

Authors have significantly improved the manuscript. However, authors still need to sufficiently present or justify that GBE50 has been sufficiently characterized as per journal requirement. Authors cited a review article (reference 50) to indicate the constituents of GBE50. Authors should rather use an original research article that has previously characterized GBE50

We look forward to receiving your revised manuscript.

Kind regards,

Tosin Abiola Olasehinde

Academic Editor

PLOS ONE

Journal Requirements:

Additional Editor Comments:

Authors have significantly improved the manuscript. However, authors still need to sufficiently present or justify that GBE50 has been sufficiently characterized as per journal requirement. Authors cited a review article (reference 50) to indicate the constituents of GBE50. Authors should rather use an original research article that has previously characterized GBE50

---

## [Author Response · Author response to Decision Letter 3]

27 Jun 2024

Editor’s Comments to the Authors:

Authors have significantly improved the manuscript. However, authors still need to sufficiently present or justify that GBE50 has been sufficiently characterized as per journal requirement. Authors cited a review article (reference 58) to indicate the constituents of GBE50. Authors should rather use an original research article that has previously characterized GBE50.

Author response: Thanks a lot. We checked the original research article that has characterized GBE50 from 2014 to the present on PubMed, and found only two such papers.

In the first article published in 2021 [1] that we cited it as Reference 21 in our manuscript, the compounds in GBE50 were identified by UPLC-Q/TOF-MS analysis. 73 compounds were identified, including terpene lactones, flavonoids and their glycosides, organic acids, flavanols and biflavones. Among them, 39 components were compared with reference substances and the relevant information was searched in databases such as PubChem and TCMSP. The related information for each component for shown in Table 1 of this article.

[1] Ke J, Li MT, Huo YJ, Cheng YQ, Guo SF, Wu Y, et al. The Synergistic Effect of Ginkgo biloba Extract 50 and Aspirin Against Platelet Aggregation. Drug Des Dev Ther. 2021;15:3543-60. doi: 10.2147/Dddt.S318515. PMID: 34429584.

In the second article published in 2023 [2], the constituents in GBE50 were analyzed by UHPLC-Q-Exactive Orbitrap HRMS. As illustrated in Table S1 of this article, altogether 38 compounds were analyzed in GBE50, including 23 flavonoids, 5 biflavonoids, 4 catechins, 5 terpene lactones, and 1 organic acid.

[2] Zhang YN, Zhu GH, Liu W, Xiong Y, Hu Q, Zhuang XY, et al. Discovery and characterization of the covalent SARS-CoV-2 3CLpro inhibitors from Ginkgo biloba extract via integrating chemoproteomic and biochemical approaches. Phytomedicine. 2023;114:154796. doi: 10.1016/j.phymed.2023.154796. PMID:37037086.

Since both articles are original research articles in which GBE50 was sufficiently characterized to indicate the class of compounds and individual components present, and the first article was cited as Reference 21 in our manuscript, we edited a part of our discussion properly and cited the second article as a new Reference 58. The detail please refer to the manuscript (with track changes in word).

Journal Requirements to the Authors:

Author response: Thank you. We double-checked our reference list to ensure that it was complete and correct, the information of Reference 2 and 7 were supplemented sufficiently, and no cited papers to date was retracted. Because a new article was citied as Reference 58, the list of Reference followed it had to be adjusted properly and did it.

---

## [Editor Report · Decision Letter 4]

11 Jul 2024

Ginkgo biloba extract inhibits hippocampal neuronal injury caused by mitochondrial oxidative stress in a rat model of Alzheimer's disease

PONE-D-23-22230R4

Dear Dr. Xia

We’re pleased to inform you that your manuscript has been judged scientifically suitable for publication and will be formally accepted for publication once it meets all outstanding technical requirements.

Kind regards,

Tosin Abiola Olasehinde

Academic Editor

PLOS ONE
---

## [Editor Report · Acceptance letter]

26 Jul 2024

PONE-D-23-22230R4 

PLOS ONE

Dear Dr. Xia, 

I'm pleased to inform you that your manuscript has been deemed suitable for publication in PLOS ONE. Congratulations! Your manuscript is now being handed over to our production team.

Kind regards, 

on behalf of

Dr. Tosin Abiola Olasehinde 

Academic Editor

PLOS ONE